# Transcranial Electrical Stimulation for Associative Memory Enhancement: State-of-the-Art from Basic to Clinical Research

**DOI:** 10.3390/life13051125

**Published:** 2023-05-02

**Authors:** Jovana Bjekić, Milica Manojlović, Saša R. Filipović

**Affiliations:** Institute for Medical Research, University of Belgrade, 11000 Belgrade, Serbia

**Keywords:** associative memory (AM), transcranial direct current stimulation (tDCS), transcranial alternating stimulation (tACS), oscillatory transcranial direct current stimulation (otDCS), transcranial electric stimulation (tES), parietal cortex, cued recall, aging, mild cognitive impairment (MCI), dementia

## Abstract

Associative memory (AM) is the ability to bind new information into complex memory representations. Noninvasive brain stimulation (NIBS), especially transcranial electric stimulation (tES), has gained increased interest in research of associative memory (AM) and its impairments. To provide an overview of the current state of knowledge, we conducted a systematic review following PRISMA guidelines covering basic and clinical research. Out of 374 identified records, 41 studies were analyzed—twenty-nine in healthy young adults, six in the aging population, three comparing older and younger adults, as well as two studies on people with MCI, and one in people with Alzheimer’s dementia. Studies using transcranial direct current stimulation (tDCS), transcranial alternating current stimulation (tACS) as well as oscillatory (otDCS) and high-definition protocols (HD-tDCS, HD-tACS) have been included. The results showed methodological heterogeneity in terms of study design, stimulation type, and parameters, as well as outcome measures. Overall, the results show that tES is a promising method for AM enhancement, especially if the stimulation is applied over the parietal cortex and the effects are assessed in cued recall paradigms.

## 1. Introduction

Human memory is one of the most powerful mental processes, which is implicated in a variety of daily experiences and activities—from remembering meaningful events to enabling goal-oriented behavior. Over the last 50 years, evidence-backed cognitive theories [1,2,3,4] categorized memory according to the duration of the storage (sensory, short- and long-term memory), modality (echoic, iconic, motor, haptic), level of awareness and consciousness involved (implicit vs explicit memory), the type of knowledge (declarative vs procedural memory, i.e., knowing *what* and knowing *how*), as well as memory domains and content (semantic, episodic, autobiographical). From a functional perspective, each type of memory can be broken down into distinct yet interrelated processes, such as encoding, retention (i.e., storage), and retrieval [5,6]. The process that plays a central role in encoding and storing complex memories and experiences is referred to as *binding* [7]. Memory binding is the function that integrates multiple elements of complex events into unified wholes. This process is at play whenever multiple items need to be stored together either for immediate manipulation (e.g., in working memory [8]) or subsequent recollection (e.g., in source memory [9]). The umbrella term for binding-dependent memories, regardless of their duration, context, modality, or domain, is associative memory (AM). 

AM represents the ability to bind previously unrelated pieces of information and store it as a unified representation that is accessible when sought for retrieval [10,11]. Therefore, AM encompasses mechanisms responsible for the formation of declarative, episodic as well as autobiographical memories, and plays an important role in day-to-day functioning. Unfortunately, AM is affected by healthy aging [12] as well as different neuropathological processes [13]. Furthermore, neuropsychological studies show that AM decline is one of the reliable indicators of cognitive impairment [13,14] and one of the prominent early signs of different types of dementia [15].

As memory deficits still do not respond well to pharmacological treatment [16], while there is evidence for their susceptibility to plasticity-based interventions (e.g., cognitive training [17]), recent years have seen an expansion of memory-oriented transcranial brain stimulation research. Transcranial brain stimulation (TBS) or noninvasive brain stimulation (NIBS) refers to a set of techniques that use different physical forces such as magnetic and electric fields, and more recently ultrasound, to harness the brain plasticity capabilities by modulating neuronal excitability and the activity of functional brain networks [18]. Here, we focus on transcranial electric stimulation (tES)—a set of NIBS techniques that use weak electrical currents (usually between 1 and 2 mA) to modulate brain activity aiming at altering behavioral responses [19].

The most used tES is bipolar transcranial direct current stimulation (tDCS), in which two electrodes of opposite polarity are placed on a person’s head [20]. The set-up in which a positively charged electrode is placed over the cortical target is referred to as anodal tDCS, whereas cathodal tDCS refers to a negatively charged electrode being placed over the target brain area [20]. Anodal tDCS is presumed to induce facilitatory effects by modulating resting state membrane potential thus increasing cortical excitability [21,22]. Unlike tDCS with constant current flow, transcranial altering stimulation (tACS) applies an oscillating current that shifts polarities between the electrodes [23,24]. These rhythmic changes in the current waveform are assumed to induce the entrainment of neural oscillations to the stimulation frequency leading to an increase in the activation of the targeted structures [25]. For a more detailed overview of the mechanisms of different tES techniques see [18,19,26].

Over the years, these two types of tES have been constantly modified and advanced to improve their effectiveness. To better target specific memory-relevant processes, custom waveforms of current delivery were created. For example, to simultaneously increase excitability and induce frequency-specific effects, oscillatory tDCS (otDCS) protocols have been developed [23]. Likewise, rhythmic stimulation with gamma bursts superimposed at the peak of theta waves to modulate theta-gamma coupling has been tried [27,28]. At the same time, to increase the focality and anatomical specificity of the stimulation, so-called high-density or high-definition (HD-tDCS, HD-tACS) stimulation set-ups were developed [29,30]. Namely, instead of using just two relatively large electrodes, the number, size, and placement of electrodes are adjusted to maximize current density at the relevant cortical region.

As the field progresses, and the new and improved tES protocols are implemented, it remains elusive which stimulation parameters contribute to the effectiveness (or lack of it) of tES for memory improvement. This review aims to fill this gap and provide an overview of the increasing number of tES studies intended to modulate AM. Therefore, we will systematize and critically evaluate the current state of knowledge with respect to study designs, tES technique (tDCS/tACS/otDCS/HD-tES), stimulation site, intensity, and other relevant parameters and outcomes (i.e., type of task and the outcome measures). We will focus on basic experimental research involving healthy human subjects and look into the attempts to apply tES in aging populations as well as clinical trials aimed at mitigating AM deficits.

## 2. Methods

The review follows Preferred Reporting Items and Systematic Review (PRISMA) guidelines [31].

### 2.1. Search Strategy and Study Selection

We searched PubMed, Scopus, and Web of Science databases using the combination of AM and tES keywords. The titles and abstracts were searched for AM-related terms (associative memory, source memory, relational memory, episodic memory, paired associate/s, learning associations, associative encoding, associative binding, face word, cued recall, word pairs), and tES-related terms (transcranial electric stimulation, tES, transcranial direct current, tDCS, transcranial altering current, tACS, HD-tDCS, HD-tACS). The exact syntax terms for each database are enclosed in Appendix A. The database search was conducted on 19 January 2023. To ensure the comprehensiveness of the review, additional records (e.g., pre-prints) were sought through a manual search of Google Scholar using the same keywords (both AM and tES-related terms) as well as the references of the articles selected from the automatic search of the databases. The search was limited to full-text original articles published in English.

### 2.2. Study Selection and Eligibility Criteria

The initial set consisted of 374 records—369 identified by database search and 5 identified manually. After removing duplicated records, 157 unique records remained. The titles and abstracts of these records were screened against the eligibility criteria. When insufficient information was provided in the abstract, the methods section of the articles was analyzed. Figure 1 presents the study selection PRISMA flow chart.

In line with our PICO strategy (Appendix B, Table A1) we included the studies with adult human participants (age ≥ 18 years), either healthy (with or without memory complaints) or with diagnosed memory deficits (e.g., mild cognitive impairment (MCI), dementia). Studies with any type of tES (tDCS, HD-tDCS, tACS, HD-tACS, otDCS) were eligible for inclusion, either having tES as a sole intervention or in combination with other memory-oriented interventions (e.g., cognitive training). At the outcome level, we included the studies that reported on the behavioral assessment of AM by either immediate or delayed cued recall, recognition, or reproduction measures. Only studies with appropriate sham-control conditions and single or double blinding procedures were included.

### 2.3. Data Extraction and Analysis

The following information was extracted from each study methods section: study population (healthy young adults/healthy aging group, clinical condition), sample size (total sample size and number of participants per group), participants age (mean age and standard deviation or range), study design (between-subjects parallel group design or within subject cross over design), stimulation type i.e., technique (tDCS, tACS, HD-tDCS, HD-tACS, otDCS), duration (minutes of active stimulation), dose (intensity in mA and frequency for oscillatory protocols), electrode positions (intended cortical target e.g., left dorsolateral prefrontal cortex (dlPFC), and electrode positions per 10-10 international EEG system e.g., F3), timing of stimulation in respect to AM assessment task (online protocols—during encoding and/or retrieval; offline protocol—before AM task), number of tES sessions (single or multiple stimulation sessions), time between the sessions, AM task and outcome measures (recognition, cued recall, reaction times, memory confidence). We also extracted reported results about the tES effects on AM-task measures.

## 3. Results

Out of the initial 374 records, after removing duplicates and excluding the studies that did not meet the inclusion criteria (e.g., were not sham-controlled, did not include associative memory measures, assessed associative memory performance only after participants slept, or were not original papers but rather metanalyses or reviews), 41 articles were included in this systematic review—29 on young healthy participants and 12 on aging and clinical population (see Figure 1). The majority of the studies (38 articles) aimed to assess the effects of tES on AM as a primary outcome measure, while three used AM as a secondary outcome.

### 3.1. Studies on Healthy Adult Participants

Most of the AM-tES research was conducted on healthy adults. A summary of the healthy-subject studies in chronological order is presented in Table 1.

Most studies (22 articles) assessed the effects of anodal tDCS, only 4 explored cathodal tDCS effects, while frequency-modulated protocols were applied in 11 studies (6 tACS and 3 otDCS). Furthermore, most experiments used bipolar 1 × 1 montage (22 studies), while 7 studies used multielectrode set-ups to deliver HD-tDCS (3 studies), HD-tACS (2 studies) or to optimize current flow to the targeted area. The current intensity was between 0.5 and 2 mA and delivered for 8–30 min. Oscillatory protocols were delivered in theta (6 studies) and gamma frequencies (2 studies) or a combination of the two [28]. Most common stimulation targets were prefrontal (PFC, 64% of studies) and parietal (30%) cortices. Thus, the target electrode was commonly positioned at either F3/F4 or P3/P4 of the 10-10 international EEG system. However, the positioning of the return electrode/s was highly inconsistent, resulting in a diverse set of montages used in the experiments. Moreover, a few studies targeted other brain areas, including the temporal cortex (e.g., [28]) and occipital nerve [58], or used unique montages targeting various brain areas at once [54]. Some versions of standard AM tasks (source memory and tasks where participants had to pair words and/or pictures) were used in all papers, along with standard outcome measures of cued recall (19 studies) or recognition (14 studies). Only a handful of papers report effects on additional measures such as subjective memory confidence (2 studies) or secondary outcome measures such as reaction time (RT) in memory tasks (2 studies).

The results mostly show significant tES effects on at least one AM outcome in young healthy participants’ experiments. Specifically, 18 out of 29 studies found positive tES effects on AM, 7 studies presented evidence of tES decreasing AM performance, while 4 studies reported null effects on all AM outcomes. There is no apparent relationship between the AM effects and the type of tES protocol applied, as positive effects were reported following anodal tDCS [32,33,44,47,49,52,53,57], tACS [58,59], otDCS [52,59], HD-tDCS [54], HD-tACS [48] and even cathodal tDCS [51].

When PPC was targeted, the effects on AM were predominantly positive [34,39,44,47,49,52,59], with only three studies showing negative [35,55] or null effects [31]. The effects of frontal stimulation were much more mixed—some experiments showed AM improvement [32,33,37,45,48,50,53], while others showed null [38,40,41,42,45,51,56] or negative effects [36,42,43]. Two-thirds of the studies that applied tES during the encoding (online protocol) showed neuromodulatory effects, while only 3 out of 7 studies that applied tES during the retrieval stage showed positive effects on AM [34,45,54]. When it comes to studies that applied tES prior to AM task (offline protocol), all but one showed a facilitatory effect. That is, AM enhancement was observed when PPC was targeted [44,47,52], and there were no effects in the study that applied tES over the premotor area [46]. Finally, with respect to the outcome measures, the studies that used cued-recall paradigms to measure AM performance showed mostly positive effects [32,33,37,44,47,52,53,54,57,58,59], while AM performance was often unaffected by tES when assessed in associative recognition paradigms [32,33,35,36,37,39,40,41,48,50,55,56].

### 3.2. Studies Conducted on Older Participants or Comparing Older vs. Younger Participants’ Effects

We found 9 studies assessing tES effects on AM in the context of aging (Table 2). Six studies were conducted on samples of older participants [61,62,63,64,65,66], aged between 53 and 90. while 3 studies compared the effects on AM performance between younger and older age groups [67,68,69].

The studies that assessed tES effects in older samples applied tDCS with standard two-electrode montage over the frontal [62,64,65], parietal [63], or temporal cortex [61]. Only one study applied tACS as well as tDCS [65]. Single-session effects were reported in 7 papers, while cumulative effects were assessed after 3 [61] and 10 sessions [64]. The follow-up assessments were present in 5 studies [61,64,66,70,71]. Beneficial tDCS effects were reported for cued recall, in one study after stimulation of the temporoparietal cortex [63], and in another that stimulated the occipital nerve [66], while the rest of the studies showed null effects [61,64,65]; one study even showed negative effects [62]. It is of note that three of the later four studies targeted the prefrontal cortex.

When it comes to comparison between young and older samples, Leach and colleagues found effects on both cued recall and recognition after applying tACS to dlPFC during encoding, but these effects were found only in the younger group [69]. In contrast, Fiori and colleagues opted for applying tDCS over Wernicke’s area during recognition and found positive effects only in the older group [67]. Lastly, Prehn et al. (2017) assessed the effects of combining tDCS with 20 mg citalopram during AM encoding and found it to be superior to solo tDCS or pharmacological treatment in both younger and older participants [68].

### 3.3. Studies on People with MCI and Alzheimer’s Disease

Two studies assessed the effects of tES on AM in people with mild cognitive impairment (MCI) and one in people diagnosed with Alzheimer’s disease (Table 2). De Sousa et al. found improved cued recall in the MCI group after applying tDCS to the temporal cortex during cognitive training [70]. However, a study that did not combine tDCS with cognitive training reported null effects in people with MCI after 5 tDCS sessions [71]. Finally, the only study that assessed Alzheimer’s patients, found that 1-hour gamma-tACS applied over PPC led to improved recognition in face-word tasks [72].

**Table 2 life-13-01125-t002:** Summary of the studies with aging and clinical samples.

Study	Sample n (Group), Age, Health Status	Design	tES ^1^	Montage	AM Task: Measures	Result
Leach et al., 2016 [62]	14 (7 + 7) 60–90 years, healthy	parallel 2-group online (encoding)	tDCS 2 mA 25 min	1 × 1 left inferior PFC (F7-arm)	face-name task: cued recall and recognition	↓ recognition (more false alarms in active group) = recall
Fiori, 2017 [67]	30 (15 + 15) young: 29 ± 6, old: 72 ± 6, healthy	parallel 2-group online (retrieval)	tDCS 2 mA 20 min	1 × 1 Wernicke’s area (CP5–CP4)	pseudowords-picture task: recognition	old: ↑ recognition young: = recognition
Prehn, 2017 [68]	40 (10 + 10 + 10 + 10) young: 18–35 years, old: 50–80 years, healthy	parallel 4-group online (encoding; w/medical intervention)	tDCS 1 mA 20 min	1 × 1 right temporoparietal cortex (T6—left frontopolar cortex)	object location task: cued recall	old: ↑ recall + medical intervention = recall young: ↑ recall + medical intervention = recall
Külzow et al., 2017 [61]	32 (16 + 16) 53–79 years, healthy	parallel 2-group offline (before encoding; w/cognitive training)	tDCS 1 mA 20 min 3 sessions	1 × 1 right temporal cortex (T6-eyebrow)	object location task: recognition (one month later)	= recognition
Antonenko et al., 2019 [63]	34 63.1 ± 7.7 years, healthy	cross-over online (encoding)	tDCS 1 mA 20 min	1 × 1 left temporoparietal cortex (CP5–AF4)	pseudowords-picture task: cued recall (immediate: 0 and 20 min)	↑ recall (in both time points)
Huo et al., 2020 [64]	49 (25 + 24) 66.6 ± 6.1 years, healthy	parallel 2-group offline	tDCS 2 mA 30 min 10 sessions	1 × 1 left dlPFC (F3—deltoid muscle)	source memory task: cued recall (24 h after last simulation)	= recall
Klink et al., 2020 [65]	28 71.1 ± 6.4 years, healthy	cross-over online (encoding)	tDCS 2 mA 20 min tACS (5 Hz) ± 1 mA 20 min	1 × 1 left vl PFC (individualized position usually between T3–F3 and F7–C3)	face occupation task: cued recall and recognition	= recall = recognition
Luckey et al., 2020 [66]	30 (15 + 15), 55–70 years, healthy	parallel 2 + group, online (encoding)	tDCS 1.5 mA 13 min	1 × 1 occipital nerve (C1–C2)	word pairs task: cued recall (immediate, 7 and 24 days later)	↑ cued recall
Leach, 2020 [69]	96 (48 + 48), young: 22.4 ± 4.7 years, old: 65.6 ± 4.9 years, healthy	parallel 2 + group online (encoding)	tACS 1.5 mA	1 × 1 left dl PFC (F3-arm)	face-name task: cued recall, recognition	old: = recall = recognition young: ↑ recall ↑ recognition
de Sousa et al., 2020 [70]	48 50–90 years, 16 MCI (8 + 8) 32 healthy (16 + 16)	parallel 4-group offline (before encoding; w/cognitive training,)	tDCS 1 mA 20 min	1 × 1 right temporal cortex (T6—supraorbital area)	object location task: cued recall (immediately and 24 h later)	MCI: ↑ cued recall (only immediate effects) healthy: = recall
Gu et al., 2022 [71]	40 (20 + 20) 64 ± 6.6 years MCI	parallel 2-group offline (before encoding)	tDCS 2 mA 20 min 5 sessions	1 × 1 left temporal area (T3-shoulder)	AM task form Wechsler Memory Scale, (immediate and 4 weeks later)	× no effects
Benussi et al., 2022 [72]	60, 72.3 ± 7.0 years Alzheimer	cross-over offline (before encoding)	tACS (40 Hz) 1.5 mA 60 min	1 × 1 PPC, precuneus (Pz—deltoid muscle)	face-name task: recognition	↑ recognition

^1^ If the number of tES sessions is not specified there was only one session or one session with each stimulation type in cross-over designs; vlPFC—the ventrolateral prefrontal cortex.

## 4. Discussion

Tackling memory decline and deficits are one of the great challenges in cognitive neuroscience and neurorehabilitation. Last several years, we are witnessing an increased interest in the application of different NIBS techniques to modulate memory. This systematic review provides insight into the state-of-the-art of applying tES to modulate AM in healthy people and clinical populations with varying levels of cognitive (memory) deficits. Most of the research involved exploring basic mechanisms in healthy adults, a few studies assessed the effects of aging, whereas clinical applications remained largely unexplored. Overall, the evidence presented here suggests that tES is a promising approach for memory enhancement, but the question of optimizing the protocols to increase effectiveness and reduce the variability of the effects is still largely unanswered. Therefore, we focus the discussion on the main challenges and highlight gaps in knowledge to be addressed in the future.

### 4.1. Defining the Optimal Stimulation Site/Were Do We Stimulate

The hippocampus and the surrounding medial temporal structures play a central role in AM [73,74], but due to their anatomical position cannot be directly modulated by tES [47]. Nonetheless, the formation and retention of memory representations are achieved through interconnectivity within a widespread hippocampo-cortical network, which includes frontal, temporal, and parietal cortices too [75]. Hence, most of the studies delivered tES to one of these cortical regions, aiming at potentially inducting network-wide effects [76].

The frontal areas, specifically left dlPFC, have been the most frequent tES target in AM studies. However, these experiments resulted in mixed findings and questionable specificity of the effects. It could be argued that, even when AM enhancement is achieved, this is conducted mostly via the facilitation of supporting processes such as attention, executive control, or reasoning that are highly dlPFC dependent [77]. On the other hand, the evidence suggests that delivering tES to the temporoparietal or posterior-parietal cortex via different tES protocols can facilitate AM [34,39,44,47,49,52,59] in a persistent and function-specific manner [44,52], even in aging samples [63,68] and persons with Alzheimer’s dementia [72]. This is in line with previous neuropsychological and neuroimaging evidence on the role of the parietal cortex, specifically PPC in memory [78,79] and in keeping up with the TMS experiments showing the functional relevance of PPC-hippocampal relay for AM [80,81,82,83]. Unfortunately, none of the studies directly compared the effects of frontal vs. parietal stimulation on AM.

Even though PPC seems as the most promising target for delivering tES, the optimal electrode set-up to do so remains elusive. There is evidence showing positive effects of the standard 1 × 1 montages, with the anode over left/right PPC and the return on the contralateral cheek (e.g., [44,59,72]), as well similar electrode placements (e.g., CP5–Fp2) [49,63]. However, alternative electrode setups such as multichannel stimulation [48] or ring electrodes [39,55] also showed memory-modulatory effects. Recent advancement in electrical field modeling allows for optimizing montages to maximize the current density in the desired brain region [84]. Such an approach has been adopted by several experiments (e.g., [39,48]), however, modeling-informed experiment focusing on PPC has not been conducted yet.

It is important to note that even when the same cortical area is targeted, a fixed electrode placement across all participants may result in variable outcomes. This may be due to individual differences in anatomy including skull characteristics, brain volume, scalp-to-cortex distances as well as overall variability in functional and structural brain properties. Moreover, almost no study has taken into account sex differences in neuroanatomical properties which might be an additional source of variability at the group level. These concerns are corroborated by studies that combine tES with different neuroimaging methods including EEG [85], MRI [86], PET [87], etc. It could be possible to account for the mentioned issues by delivering individual-level neuroimaging-guided tES with electrodes placed and intensities adjusted based on the current modeling for each participant. However, this approach has not yet been implemented in tES-AM studies, therefore its incremental value in reducing variability is yet to be evaluated.

### 4.2. Stimulation Protocol/How Do We Stimulate

Although all reviewed tES studies on AM applied low-intensity current to modulate brain activity, protocols differed in intensity (dose) and waveform of the current applied. The current intensity was in all studies between 0.5 and 2 mA, which is within recommended safety limits [88]. However, the selected intensities were rarely justified and discussed in the papers. Only one study compared the effects of different stimulation intensities (1 mA vs. 1.5 mA) and found that only 1.5 mA had significant effects on AM [53]. Therefore, in light of the evidence showing a non-linear relationship between current intensity and physiological effects [89], it is difficult to draw conclusions about optimal dosing. Selecting appropriate stimulation intensity is in general an open question in NIBS-based neuromodulation of non-motor cortical areas where there is no direct physiological readout, which could guide the dosing. This is particularly emphasized in the tES application where, in contrast to transcranial magnetic stimulation (TMS), not even a threshold intensity for the motor cortex is available. Moreover, since the current density in the brain tissue depends on individual neuroanatomy, there are strong arguments to move towards individualized dosing [90,91,92]. That is, making sure that the current density in the targeted brain region is equal across all participants, rather than applying the same intensity to all participants [91,92].

As a rule, early NIBS studies of AM modulation applied constant anodal tDCS [32,33,34,35]. This type of tES is expected to increase the excitability of the cortical tissue under the positively charged electrode, and have facilitatory effects on cognitive performance—in this case AM. Nevertheless, tDCS has low spatial focality [93], thus the specificity of the effects is highly questionable, especially if the stimulation is delivered to dlPFC and no control-function tasks are included in the study design. Despite some studies showing evidence of function-specific tDCS effects when applied over PPC (e.g., [47,49]), the induced electric fields are widely distributed across different cortical regions. The poor spatial resolution of the classic tES techniques led to the development of the tES techniques aiming at higher focality, such as HD-tDCS [30]. Unfortunately, the first few studies that applied HD-tDCS did not provide convincing evidence for AM neuromodulation [38,39].

The path towards increased specificity of the effects opened with the application of tACS. This type of tES generates oscillating electrical fields that can modulate brain rhythms underlying targeted function causing thus a change in performance [24]. The first tACS study on AM compared the effects of combined theta (5 Hz) and high gamma (80 Hz) frequency stimulation. Namely, across 3 experiments de Lara and collages assessed the effects of sinusoidal stimulation with gamma bursts at peaks or troughs of the theta wave, or continuous gamma-oscillations superimposed on theta frequency [28]. They found high inter-individual variability in the effects, but gamma bursts at the troughs led to a reduced cued recall. The tACS studies that followed resulted in mixed findings [50,58,59]. However, the only study aimed at the clinical application of tACS resulted in AM improvement in a group of people with Alzheimer’s dementia [72]. In an interesting attempt to combine increased focality and specificity of stimulation Lang and colleagues [48] delivered HD-tACS (6 Hz) over dlPFC and showed its superior effects in comparison to HD-tDCS delivered using the same electrode montage. However, the theoretically expected superiority of tACS over tDCS was not conceptually replicated in the follow-up studies [55].

As tDCS and tACS employ different, yet not mutually exclusive mechanisms to modulate brain activity, they can be combined to deliver so-called otDCS [23]. In this technique, the current oscillates within the same polarity, which is presumed to induce modulation of brain rhythms at the state of increased excitability. Relying on the relevance of theta-band activity for AM [94,95], the studies that applied theta-otDCS showed impaired cued call when electrodes were positioned over the frontal [43] and improvement when placed over the parietal cortex [52,59]. In a recent study, Živanović et al. [59] comparatively assessed the effects of anodal tDCS, theta-band tACS, and theta-band otDCS, and found all protocols to have facilitatory effects. Moreover, it shed light on how different modes of action can affect AM at different levels of task difficulty—with oscillatory protocols being more effective when the memory load was higher [59]. Therefore, it seems that complementary modes of action, i.e., increased excitability of the relevant cortical regions coupled with network-wide oscillatory entrainment, can be beneficial in promoting mnemonic functions.

### 4.3. Timing of Stimulation and the Duration of the Effects/When Do We Stimulate and How Often

To modulate cognition tES is either applied in so-called online or offline protocols, that is—either during or before the task [19]. The effects registered in these two types of protocols provide evidence for different neurophysiological changes induced by tES [21]. Namely, the effects during the tES stimulation (online) are dependent either on the changes in membrane potential altering neuronal excitability and modulating response to the incoming signals in tDCS, or changes in the spontaneous function-related neuronal oscillatory activity in tACS, or on a combination of both in otDCS [59]. In contrast, the effects after the tES (offline) are supposed to be mostly driven by LTP-like changes in synaptic strength within relevant functional networks [21].

As for online protocol, in this review, we found 24 studies applying the tES during encoding, 7 studies during the retrieval phase, and, interestingly, none during both phases of the AM task. Applying tES during the encoding carries the implicit assumption that the stimulation will facilitate the binding process and that the storing of so-acquired memories will be deeper and more successful which in turn will result in better retrieval, whereas applying during retrieval could be expected to facilitate access to the stored information. On direct comparison, studies that applied tES during encoding have been more likely to modulate AM. This is in keeping with the idea that tES-induced neuromodulation is affecting the binding which is the central component of AM.

Still, exploring if stimulation over a certain brain area facilitates encoding or retrieval (or both), is an interesting one. However, the successfulness of encoding cannot be measured independently of retrieval on the behavioral level, thus this question remains to be addressed by combining behavioral and neurophysiological/neuroimaging data.

In addition to enabling us to gain a better understanding of how tES affects different memory processes, online protocols have limited potential when it comes to translating basic NIBS research into aging and clinical applications. To reduce memory deficits, plasticity-inducing offline protocols seem like a more obvious choice. There is evidence that applying tES for 20 min in healthy [44,49,52,59] and people with MCI [70], or even 1 h in people with Alzheimer’s [72] can lead to better subsequent AM performance. Still, a single tES session might not be enough to induce lasting behavioral changes (e.g., [42,51]). Repeated administration in multiple sessions over several days could be expected to lead to more consistent facilitatory effects, as was the case in one of the first AM-tES studies [32]. However, studies in older adults and people with MCI did not show AM improvement after 3, 5, or even 10 sessions [61,64,71]. Even so, due to a small number of multiple-session studies, it is difficult to pinpoint if the null results were the consequence of stimulation site, dose, duration, or other tES parameters.

It also remains unclear what would be the optimal number of sessions and the time between them for inducing even short-to-midterm lasting changes. The guidance for this can be sought in the studies that included follow-up AM assessment. Namely, experiments on healthy adults showed that the effects of a single tES session could still be observed 24 h later [37,44,44,52,53], while the evidence for 5- or 7-days aftereffects although present is much less convincing [44,52,57,66]. Thus, the practice adopted in the multi-session studies so far, to separate sessions between one and a few days, seems as a reasonable approach for further clinically oriented studies.

### 4.4. Outcome Measures

AM can be assessed using different behavioral paradigms. The vast majority of tES studies used either cued recall or associative recognition. After learning new associations in the cued recall paradigm, participants are presented with one piece of information (cue), and their task is to recall the item or context it was associated with, while in the associative recognition paradigm, both parts of information are presented in either correct (i.e., same as in learning phase) or recombined manner. This review shows that tES effects are more likely to be detected when cued recall paradigm is used to assess AM [32,33,37,44,47,52,53,54,57,58,59]. There are several reasons why this might be the case. Although both paradigms are highly dependent on the successfulness of encoding, recognition is less demanding at the retrieval stage. That is, even when equally well encoded, one can fail to recall the information when prompted with the cue, while still being able to correctly recognize it when being shown the unified representation. In keeping up with this, cued recall is often more challenging for participants than recognition, which makes it from a psychometric perspective more sensitive to detect small-size tES effects. Moreover, it could be argued that cued recall is a more focal measure of AM, since in recognition paradigms other processes, such as probability-based decision-making and response style, strategies, and biases, play a significant role too [96].

In addition to main AM outcomes (i.e., % of correctly recalled items or recognized pairs), some studies reported on tES effects on other measures, such as reaction times [39,56] or memory confidence [35,50]. Even though these results do not necessarily provide evidence of AM modulation, they provide insight into how tES affects memory functions. For example, shorter reaction times for correctly recognized pairs [39,56], might point towards tES-facilitated quicker (i.e., easier) access to memory representations. Similarly, increased memory confidence when coupled with better performance might suggest more prominent AM representations.

### 4.5. Methodological Concerns: Sample Size, Power Issues, and Blinding

To assess tES effects on AM against sham-control conditions, studies adopt either parallel-group (where one group of participants receives sham and one or more groups receive real tES) or cross-over designs (where the same group of participants undergoes both sham and real tES in counterbalanced order). From a statistical perspective, the total number of participants enrolled in the study does not directly translate into the statistical power; what is more important is the number of observations per condition. This review shows that regardless of the study design researchers opt for a similar number of participants (i.e., observations) per condition, usually between 15 and 25.

This might be one of the hidden sources of variability in the presented findings. Namely, similar sample sizes across different study designs result in very different power, thus different effect sizes are needed to show statistically significant effects. That is, with the same number of observations per condition, to reach the statistical significance threshold (*p* = 0.05), tES effects need to be almost 30% larger in the two-group design than in the crossover design (for n = 20 and the power of 0.80; d = 0.91 vs. d = 0.66). With that in mind, it is difficult to say if the null findings presented in the studies with 7, 13, or 15 participants per group [33,35,38,54,56,62,67] are simply the result of insufficient power, or if the facilitatory effects in low-powered studies are type-1 error. On the positive note, higher-powered studies showed modulatory effects of tES on AM in healthy [52,53,59], aging [63], and clinical samples [70,72], which allows for better estimation of the expected effect sizes and more data-driven determination of the sample sizes in the future.

To adequately address this issue, the researchers should, whenever it is possible, rely on *a priori* power calculations to determine the adequate sample size for their study. Even when this was not carried out prior to the data collection, it is useful to report on achieved power, so that the reported effects or the absence of them could be interpreted in the appropriate manner (see [48] for example).

Another important issue that might contribute to the variability of the effects is the sham protocol used, its features, and the effectiveness of participants’ blinding (i.e., the ability of the participants to distinguish between real and sham stimulation). In the reviewed studies different types of sham protocols were used. The most frequently used approach involved short ramp-up/down periods at the beginning and the very end of the stimulation [31,33,35,37,39,40,43,44,45,46,47,51,52,55,58,71]. Some researchers opted for a sham protocol with a ramp-up/down only at the beginning of the stimulation [28,43,49,55,65,66,68,71]. However, there were studies applying the same current intensity as in the active stimulation protocol, but for a brief period of time—usually 30 s at the beginning of stimulation [35,39,51,53,57,58,61,64,70]. Lastly, a handful of studies [33,62,69] applied low-intensity current (0.1 mA) throughout the stimulation period, as such a low intensity should not have physiological effects but could still induce some skin sensations. It is difficult to say which of these sham protocols is most effective, as only three studies [52,59,69] reported on the actual effectiveness of sham blinding. All of them showed that the ability to guess when the sham protocol was administered did not affect associative memory performance. 

Although it is highly desirable that future studies report the data on the effectiveness of sham control, a recent study suggests that participants’ beliefs and expectations about the stimulation (active vs sham) do not moderate tDCS effects on memory [97]. Namely, Stanković and colleagues analyzed data from over 200 tDCS sessions and found a lack of placebo-like effects stemming from participants’ beliefs about the stimulation type they received—the participants’ beliefs did not influence the performance on associative and working memory tasks [97].

## 5. Conclusions

This systematic review of the current state of the literature focused on the application of different tES techniques to modulate AM in healthy, aging, and clinical populations. In search for the optimal tES technique and protocol to induce meaningful changes in memory performance, we found that studies that reported the strongest effects on AM tend to stimulate over the parietal lobe and use cued recall paradigms. So far, there is more evidence of tDCS effectiveness than other tES techniques. However, this is mainly due to the high prevalence in usage and there is a need to further examine the effects of tACS and otDCS on AM, especially on aging and individuals with memory deficits. Similarly, although we found that online protocols with active stimulation during encoding tend to be effectual, there is sufficient evidence that offline protocols stimulating before encoding are equivalently effective. Further empirical studies focusing on the systematic comparisons between different stimulation protocols and their specific features are needed before translating healthy participants’ findings into clinical applications.

## Figures and Tables

**Figure 1 life-13-01125-f001:**
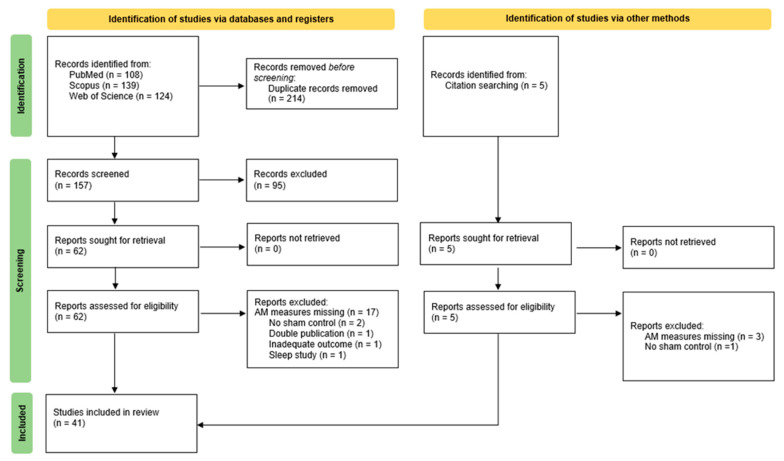
The PRISMA flow chart.

**Table 1 life-13-01125-t001:** Summary of the studies conducted on healthy young adults.

Study	Sample n (Group), Age	Design	tES	Montage	AM Task: Measures	Result
Meinzer, 2014 [32]	n = 40 (20 + 20) 23.9 ± 3.6 years	parallel 2-group online (encoding)	tDCS 1 mA 20 min 5 sessions	1 × 1 left PFC (CP5-supraorbital)	object-nonword task: cued recall, recognition	↑ cued recall = recognition
Matzen, 2015 [33]	n = 26 (13 + 13) 19-30 years	parallel 2-group online (encoding)	tDCS 2 mA 30 min	1 × 1 left PFC (F9-arm)	face-word task: cued recall, recognition	↑ cued recall = recognition
Pergolizzi, 2016 [34]	n = 54 (18 + 18 + 18) 19.6 ± 3.06 years	parallel 3-group online (retrieval)	tDCS 2 mA 20 min	1 × 1 left PPC (CP3-CP4) left dlPFC (F3-F4)	source memory task: cued recall, source bias	↓ source bias = cued recall
Chen, 2016 [35]	n = 36 (18 + 18), 21.2 years	mixed parallel 2-group: anodal/cathodal, repeated: stimulation site online (retrieval)	tDCS anodal/cathodal +1.5 mA/−1.5 mA 10 min	1 × 1 left PPC (P3-cheek) right M1 (M1-cheek)	source memory task: recognition, memory confidence	anodal tDCS: = recognition (PPC, M1) = confidence (PPC, M1) cathodal tDCS: ↓ recognition (PPC) = recognition (M1) = confidence (PPC, M1)
Gaynor, 2017 [36]	n = 72 (24), 20.8 ± 3.3 years	parallel 3-group online (encoding)	tDCS 2 mA 20 min	1 × 1 left PPC (CP3-CP4) left dlPFC (F3—supraorbital bridge)	unrelated word pairs: recognition (24 h later)	↓ recognition (dlPFC) = recognition (PPC)
Leshikar 2017 [37]	n = 42 (21) 22.5 years	parallel 2-group online (encoding)	tDCS 1.5 mA 25 min	1 × 1 left dlPFC (F3-arm)	face-name task: cued recall, recognition	↑ cued recall = recognition
de Lara, 2017 [38]	n = 30 (15), 24.8 ± 3.5 years	mixed parallel 2-group: encoding/retrieval repeated: active/sham online (encoding/retrieval)	HD-tDCS 1 mA 20 min	1 × 4 left dlPFC (AF3—6 cm distance from AF3 and 10 cm between each other)	semantically related word pairs: cued recall	= cued recall
Perceval, 2017 [39]	n = 50 (25), 23.16 ± 3.79 years	parallel 2-group online (encoding)	HD-tDCS 1 mA 20 min	ring electrode left temporoparietal cortex (CP5)	pseudoword-picture task: recognition	= recognition ↑ RT for correct pairs
de Lara, 2018 [28]	n = 24 23.5 ± 3.1 years	cross-over, online (encoding)	tACS (5 Hz + 80 Hz at peaks) 1 mA 10 min	1 × 2 left temporal lobe (T7–T8 and FPz)	semantically related word pairs, cued recall	= cued recall
	n = 24 24.3 ± 2.9 years	cross-over, online (encoding)	tACS (5 Hz + 80 Hz at troughs) 1 mA 10 min	1 × 2 left temporal lobe (T7–T8 and FPz)	semantically related word pairs, cued recall	↓ cued recall
	n = 24 23.2 ± 2.2 years	cross-over, online (encoding)	tACS (5 Hz + 80 Hz throughout) 1 mA 10 min	1 × 2 left temporal lobe (T7–T8 and FPz)	semantically related word pairs, cued recall	= cued recall
Brunye, 2018 [40]	n = 50 (25) 22.5 years	parallel 2-group online (encoding)	tDCS 1.5 mA 20 min	2 × 3 PFC (FP1, FP2–AF3, F4, P8)	face-picture task, recognition	= recognition
Moseley, 2018 [41]	n = 36 20.14 ± 2.5 years	cross-over, online (encoding)	tDCS 1.5 mA 15 min	1 × 1 right amPFC—left STG; right amPFC—left V5	source memory task: recognition	= recognition
	n = 36 22.69 ± 5.7 years	cross-over, online (retrieval)	tDCS 1.5 mA 15 min	1 × 1 right amPFC—left STG; right amPFC—left V5	source memory task: recognition	= recognition
Marián, 2018 [42]	n = 66 (33) 23.2 ± 2.5 years	parallel 2-group offline (consolidation)	tDCS 2 mA 15 min	1 × 1 right dlPFC (F4–Cz)	word pairs task: cued recall	↓ cued recall
	n = 52 (26) 23.2 ± 2.5 years	parallel 2-group offline (consolidation)	tDCS 2 mA 15 min	1 × 1 right dlPFC (F4–Cz)	word pairs task: cued recall	= cued recall
Mizrak, 2018 [43]	n = 21 /	cross-over offline (consolidation)	otDCS (5.5 Hz) 0.5–1 mA 20 min	1 × 1 left dlPFC (F3—supraorbital area)	source memory task: cued recall	↓ cued recall
Bjekić, 2019 [44]	n = 37 25.34 ± 3.59 years	cross-over offline (before encoding)	tDCS 1.5 mA 20 min	1 × 1 left PPC (P3–cheek)	face word task: cued recall	↑ cued recall
Westphal, 2019 [45]	n = 72 (24) 20 years	parallel -3 group online (retrieval)	tDCS anodal/cathodal 1.5 mA/−1.5 mA 30 min	1 × 1 left rtPFC (midpoint between FP1 and F7–C4)	source memory task: recognition	anodal tDCS: ↑ recognition cathodal tDCS: = recognition
Leclerc, 2019 [46]	n = 48 (23/25) 24.77 ± 5.33 years	parallel 2-group offline (before encoding)	tDCS 2 mA 20 min	1 × 1 premotor cortex (Fz–deltoid)	source memory task: cued recall	= cued recall
Bjekić, 2019 [47]	n = 20 26.4 ± 3.71 years	cross-over offline (before encoding)	tDCS 1.5 mA 20 min	1 × 1 left PPC (P3 -cheek)	face word task: cued recall	↑ cued recall
	n = 21 24.15 ± 2.74 years	cross-over offline (before encoding)	tDCS 1.5 mA 20 min	1 × 1 right PPC (P4-cheek)	object location task: cued recall	↑ cued recall
Lang, 2019 [48]	n = 59 (19/21/19) 18–45 years	parallel 3-group online (encoding)	HD-tACS (6 Hz)/HD-tDCS 2 mA 10 min	1 × 4 dlPFC (P10–FP1, P2, P3, PO7)	face scene task: recognition	hd tACS: ↑ recognition hd tDCS: = recognition
Owusu, 2020 [49]	n = 20 21–34 years	cross-over online (encoding)	tDCS 1.5 mA 20 min	1 × 1 left posterior temporoparietal junction (CP5–FP2)	polysemous words and meaning task: cued recall and recognition	↑ recognition ↑ cued recall
Ergo, 2020 [50]	n = 76 (38) 20.8 ± 2.4 years	parallel 2-group online (encoding)	tACS (6 Hz) 2 mA 16.6 min	1 × 1 dlPFC (FCz-neck)	word pairs task: recognition, memory confidence	= recognition ↑ confidence in correct recognition
Gilson, 2021 [51]	n = 69 (16/16/17/20) 22.4 years	parallel 4-group offline (consolidation)	tDCS anodal/cathodal with or without cognitive training 1 mA 20 min	1 × 1 dlPFC (F3–F4)	affective word pairs task: cued recall	anodal + training: ↑ cued-recall cathodal + training: ↑ cued-recall anodal: = cued recall cathodal: = cued recall
Vulić, 2021 [52]	n = 36; 18 follow up 23.8 ± 1.8 years	cross-over offline (before task)	tDCS, otDCS (5 Hz) 1.5 mA 20 min	1 × 1 left PPC (P3-cheek)	face word pairs task, cued recall	tDCS: ↑ cued recall otDCS: ↑ cued recall
Bolling, 2021 [53]	n = 56 (34/41/39) 18-25 years	parallel 3-group online (encoding)	tDCS 1.5 mA vs. 1 mA 20 min	1 × 1 dlPFC (F3–F4)	word pairs task, cued recall	↑ cued recall (1.5 mA) = cued recall (1 mA)
Huang, 2021 [54]	n = 84 (aDMN:13/14/13 pDMN:16/11/17) 19 ± 1.2 years	parallel 6-group online (retrieval)	HD-tDCS 1 mA 10 min	1 × 3 aDMN (FPz–Fz, FP1, FP2) pDMN (Pz–Oz, PO7, PO8)	word pairs task: cued recall	↑ recall in HD-tDCS anodal stimulation on pDMN ↑ recall in HD-tDCS cathodal stimulation on aDMN
Meng, 2021 [55]	n = 20 21.7 ± 8.2 years	cross-over online (encoding)	HD-tACS (6 Hz) 2 mA 15 min	ring electrode left PPC (P3)	face scene task: recognition	↓ recognition
Fernández, 2021 [56]	n = 30 (15) 21.3 years	parallel 2-group online (encoding)	tDCS 2 mA 18 min	1 × 1 dlPFC (F3–FP2)	word pairs task: recognition	= recognition ↑ RT for correct 24 h later
Pyke, 2021 [57]	n = 25 19.2 ± 0.8 years	cross-over online (encoding)	tDCS 1.5 mA 15 min	1 × 1 dlPFC (F3-wrist)	word picture pairs task: cued recall	↑ cued recall
Luckey, 2022 [58]	n = 84 (25 + 25 + 24 + 11) 21.6 ± 2.1 years	parallel 4-group online (encoding)	tDCS tACS (40 Hz) tACS (1 Hz) 1.5 mA 8 min	1 × 1 occipital nerve (occipital lobe)	word pairs task: cued recall	tDCS: = cued recall (immediate) ↑ cued recall (delayed) tACS (40 Hz): ↑ cued recall (immediate) ↑ cued recall (delayed) tACS (1 Hz): = cued recall (immediate & delayed)
Živanović, 2022 [59]	n = 40 25.2 ± 3.7 years	cross-over online (encoding)	tDCS 1.5 mA 20 min tACS (ITF) ±1 mA 20 min otDCS (ITF) 1 mA–2 mA 20 min	1 × 1 left PPC (P3-cheek)	short-term AM task number-color pairs, cued recall	tDCS: ↑ cued recall tACS: ↑ cued recall otDCS: ↑ cued recall

Note: Sample size is presented as the total number of participants included in the study as well as the number of participants per group in a parallel group design. The study designs are labeled parallel-group, crossover, or mixed to indicate between-subjects, within-subjects comparisons, or the presence of both repeated and not repeated factors. For online protocols, we indicate the phase of the task in which the stimulation was delivered. For tDCS studies, the minus sign before intensity represents cathodal stimulation; for tACS studies, intensity is presented as peak-to-peak. For tES montages 1 × 1, 1 × 4, and 2 × 3 show the number of electrodes used in the setup; return electrode(s) is always presented second. The results always show the comparison of active against sham condition/group. rlPFC—rostrolateral prefrontal cortex; aDMN—anterior default mode network; pDMN—posterior default mode network; ITF—individual theta frequency. The studies were quite diverse with respect to methods and designs. Namely, the sample sizes were between 15 and 40 in within-subject crossover designs (12 studies), and between 26 and 84 in parallel group designs (17 studies), with 11 to 41 participants per group—for power estimates see [60]. The effects were assessed in *online* protocols where tES was delivered during encoding (18 studies) or retrieval (6 studies), as well as in *offline* protocols where tES was applied before AM task (4 studies), or during consolidation i.e., between encoding and retrieval (3 studies). All but one study [36] reported immediate effects of tES on AM, with 14 studies reporting on follow-up assessment after 24 h [37,42,44,47,48,52,53,56,57], 5 days [44,52] or 7 days [51,53,57,58]. The cumulative effects of multi-day stimulation were assessed in one study [32], while three studies combined tES with cognitive training [50,51,58].

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
