# Peer review of "Transcranial Electrical Stimulation for Associative Memory Enhancement: State-of-the-Art from Basic to Clinical Research"

_life, 2023, doi:10.3390/life13051125_

Round 1

Reviewer 1 Report

This is a very interesting and well written review about an important topic. I have some comments and suggestions which may improve the quality of this paper.

1. Please address the issue of effective blinding. This reviewer thinks that placebo effects play a dominant role in tDCS studies.

2. Expectations and beliefs of research subjects influence the outcome. Any study considered that?

3. Variability of outcomes is a major concern. Please address sex, skull characteristics, brain volume, brain metabolites, and differences in scalp to cortex distances between male and female.

4. More information is needed about the underlying mechanisms of non- invasive brain stimulation. For example, the role of PET in tDCS research (Rudroff et al. 2020, Brain Sciences). Mechanisms are ignored in this review. At least, it should be mentioned in the conclusions.

Author Response

Thank you for your positive evaluation of our paper and thoughtful comments and suggestions that helped us improve our work. Please find attached the point-by-point response. 

Reviewer 2 Report

The manuscript titled “Transcranial electrical stimulation for associative memory enhancement: State-of-art from basic to clinical research” proposes a systematic review on the use of transcranial Electrical Stimulation (tES) for Associative Memory (AM). Authors analysed 41 studies and provided a critical overview on the current state of knowledge about the topic.

I carefully read the manuscript and I suppose it may be of interest for readers of Life. Nevertheless, it could be worth considering some points before the publication. Below there are my comments and suggestions.

Introduction

Overall, in the introduction section Authors accounted for their study with respect of previous research on tED and AM. They supported their purpose with appropriate available references and the rationale of the study is clear as well. The aims of the study may be better detailed, taking into account the novelty of the systematic review.

Methods

Search strategy and study selection

As suggested by PRISMA guidelines, the purpose of the systematic review should be specified with the PICO strategy. Furthermore, Authors should report why they have searched studies on PubMed, Scopus and Web of Science and not on other scientific databases. Lastly, to ensure the reproducibility of the study the exact research string with syntax terms should be reported.

Study selection and eligibility criteria

Authors claim that 5 records were identified manually. Could they specify and detail it process?

Furthermore, the PRISMA flow chart is not clear: 371 records were identified from Pubmed, Scopus and Web of Science and then 218 duplicate records were removed. Records screened after this process should be 153.

Did Authors perform a quality assessment procedure on the selected studies, as suggested by PRISMA guidelines? It would be expected to be done and reported.

Results

The identification and screening steps should be detailed also in the text. Overall, results were clearly described and reported in tables.

Discussion

At the beginning of the discussion section may be useful to provide a brief recap of the study, coherently with its aims.

Author Response

Thank you for your positive evaluation of our paper and thoughtful comments and suggestions that helped us improve our work. Please find attached the point-by-point response to comments made by both you and Reviewer no 1

Round 2

Reviewer 1 Report

Thank you for your responses to my comments and suggestions. Nice work.

Reviewer 2 Report

I carefully read the revised version of the manuscript, and I think that Authors have properly addressed all the issues raised by the reviewers.